# Dielectric Thermal Smart Glass Based on Tunable Helical Polymer-Based Superstructure for Biosensor with Antibacterial Property

**DOI:** 10.3390/polym13020245

**Published:** 2021-01-13

**Authors:** Haw-Ming Huang, Fu-Lun Chen, Ping-Yuan Lin, Yu-Cheng Hsiao

**Affiliations:** 1School of Dentistry, College of Oral Medicine, Taipei Medical University, Taipei 11031, Taiwan; hhm@tmu.edu.tw; 2Graduate Institute of Biomedical Optomechatronics, College of Biomedical Engineering, Taipei Medical University, Taipei 11031, Taiwan; duncanlin123@gmail.com; 3Department of Internal Medicine, Division of Infectious Diseases, Taipei Municipal Wan Fang Hospital, Taipei Medical University, Taipei 11031, Taiwan; 96003@w.tmu.edu.tw; 4Department of Internal Medicine, School of Medicine, College of Medicine, Taipei Medical University, Taipei 11031, Taiwan; 5International PhD Program for Biomedical Engineering, Taipei Medical University, Taipei 11031, Taiwan; 6Cell Physiology and Molecular Image Research Center, Taipei Municipal Wan Fang Hospital, Taipei Medical University, Taipei 11031, Taiwan

**Keywords:** cholesteric liquid crystals, label-free biosensor, dielectric heating

## Abstract

A dielectric thermal smart glass (DTSG) based on the dielectric heating optical (DHO) effect in tunable helical polymer-based superstructures—cholesteric liquid crystals (CLCs)—was exhibited in this study. Field-induced dielectric heating can strongly affect the orientation of liquid crystals and change its optical properties. The purpose of this research focuses on dual-frequency CLC materials characterized by their specific properties on dielectric relaxation and demonstrates their potential for antibacterial biosensor applications. The developed DTSG is driven by voltages with modulated frequencies. The principal of DTSG in transparent states are a planar (P) state and a heated planar (HP) state reflecting infrared light, operated with the voltage at low and high frequencies, respectively. The scattering states are a focal conic (FC) state and a heated FC (HFC) state, with an applied frequency near the crossover frequency. The biomolecule detection of the antibacterial property was also demonstrated. The detection limitation of the DTSG biosensor was found to be about 0.5 µg/mL. The DTSG material has many potential industrial applications, such as in buildings, photonic devices, and biosensor applications.

## 1. Introduction

Smart glass (SG), which can dynamically alter the transmittance of light by applying an electric field, has attracted much attention [1,2]. Liquid crystal (LC) materials are the most important technology for SG applications. Optical states of liquid crystal smart glass (LCSG) are typically a transparent state and a scattering state, and one can switch between them by applying different voltages [3]. There are vast applications of LCSG, such as in buildings, displays, cars, signboards, and airplanes. Typically, there are three types of commercial LC-based SG: polymer-dispersed liquid crystal (PDLC), polymer-stabilized cholesteric texture (PSCT), and polymer-stabilized liquid crystal (PSLC). Recently, applications based on dual-frequency cholesteric liquid crystals (DFCLCs) as bistable and multistable optical devices have also been demonstrated [4,5,6]. Rapid and direct switching between transparent and scattering states has previously been shown in DFCLCs [5]. The relaxation frequency characteristics of DFCLCs depending on a unique dielectric heating or thermodielectric effect have also been reported [7,8]. This thermodielectric effect is caused by the rotation of the LC molecular dipole within the dielectric material.

In general, the dielectric relaxation of LCs occurs at a very high frequency (>MHz), regardless of the electric constants ε_||_ and ε_⊥_, along the long and short axis of LC molecules. However, dielectric relaxation of dual-frequency liquid crystals (DFLCs) parallel to the long axis of LC molecules can occur at a lower frequency range, that is, between 10 and 100 kHz. Thus, it is easier to observe the dielectric relaxation frequency of orientation in dual-frequency nematic liquid crystals (DFNLCs) or DFCLCs. The dipole of DFLC materials will rotate with an applied electrical field, and then result in apparent heat. Thus, the thermodielectric effect can be easily monitored in voltage-driven DFLCs or DFCLCs because of the relatively low relaxation frequency of long axis dielectric relaxation. Based on the exceptional dielectric heating or thermodielectric effect in LCs, many applications have been proposed, such as low-voltage devices [7], photonic crystal devices [8,9,10], and electrically tunable devices [11,12]. Compared to the well-known electro-optical (EO) effects of LCs, the dielectric heating optical (DHO) effect possesses distinctive optical features. Furthermore, in DFLCs or DFCLCs, the dielectric anisotropy (Δε = ε_||_ − ε_⊥_) reverses its sign from positive to negative at a particular frequency called the crossover frequency (*f_c_*) [13]. With this specific feature, DFLCs and DFCLCs have a high potential for fast switchable and multi-functional optical devices.

The previous LC-based SGs only emphasize the electrical tuning optical features. Temperature adjustment is also a key function to be considered for the development of SG. However, an LC-based SG with adjustable transmittance and heating ability has not been invented until now. In this study, a novel dielectric thermal smart glass (DTSG) was first developed using the DHO effect in composite DFCLCs. Characteristics of the DTSG, including temperature control, frequency modulation, and optical stability, have been experimentally demonstrated. Compared to previously reported LCSGs, our proposed DTSG can rapidly switch to the heated states, heated focal conic (HFC) state, and heated planar (HP) state by applying a high-frequency electric field. When a low-frequency electric field was applied to the DTSG, the device returned to its room temperature. Thus, exhibited the scattered focal conic (FC) state and transparent planar (P) state, defined as cold FC state and cold P state, respectively. Figure 1 shows the schematic of both cold and heated modes of the DTSG device and the operation principle. The operation and DHO property of the DTSG was employed to detect Bovine serum albumin (BSA) with antibacterial possibility.

## 2. Materials and Methods

### 2.1. Materials of DTSG

The host material used for preparing the DFCLC was nematic LC, MLC-2048 (Merck, Fort Kennerworth, NJ, USA), with the clearing point 106.2 °C and *f_c_* = 14 kHz at 25 °C. The chiral dopant S811 (DIC) was doped into the host material at a concentration of 5 wt%. The helical twisting power (HTP) of S811 in MLC-2048 is −14 µm^−1^. The composite of the DFCLC materials was injected into 10 µm thick cells with anti-parallel alignment layers by capillary action. The alignment layers forced the DFCLC to exhibit an initial planar state. The frequency-dependent light transmission was examined through a probe laser beam, derived from a He–Ne laser system operated at a wavelength of 632.8 nm. Moreover, crossed polarizer schemes were adopted for the electro-optical measurements of the DTSG device. An arbitrary function generator (Tektronix AFG-3022B, Beaverton, OR, USA) was employed to apply frequency-modulated voltages. In addition, transmission spectra of the DTSG were obtained with a fiber-optic spectrometer (Ocean Optics HR 2000+, Orlando, FL, USA) with a halogen light source. All experimental data were acquired at 26 ± 1 °C.

### 2.2. Fabrication Methods of DTSG Biosensor

To prepare the DTSG biosensor for detecting BSA, the Indium Tin Oxide; ITO glass substrate was first immersed in a liquid form solution for 0.5 h, containing 1.6% DMOAP (1-octadecanaminium, and N,N-dimethyl-N-[3-(trimethoxysilyl)propyl] chloride) in deionized water, at 25 °C. In the BSA immobilization experiment, the 0.5–10 μg/mL BSA solution was immersed on the DMOAP substrate for 30 min. After being rinsed with deionized water to remove unbound BSA, the sandwich DTSG biosensor with two DMOAP coated substrates was used. In addition, the 10 μm spacers were used to form spaces in the cell. Then, the DFCLC material was employed to fill the DTSG biosensor by capillary action.

## 3. Results and Discussions

### 3.1. Mechanisms of the DTSG Device

Figure 1 schematically shows transmission dependent on the applied frequency in the DTSG device. The optical states of the DTSG system were the P, FC, HFC, and HP states with increasing frequency of the applied voltage. The P and HP states were transparent in visible spectral range, while FC and HFC states appeared opaque, caused by light scattering. Planar structures in the DTSG device were in the low-frequency range (<10 kHz). The helical axis of the transparent P state was normal to the cell substrates, and the pitch length was designed to reflect light in the near-infrared selectively. When the applied frequency was increased, the scattering FC state appeared. However, the dielectric heating effect was generated when the applied electrical frequency to the *f_c_* was increased. Dielectric dispersion in DFCLCs caused a remarkable increase in temperature because of the dielectric loss in the long axis of LCs [14]. The FC state transformed into the HFC state at a higher temperature when the applied frequency exceeded *f_c_*. If the applied frequency is increased continuously, the negative dielectric anisotropic Δ*ε* becomes dominant, and the transparent HP state occurred. Figure 2a compares the frequency response transmittance at 70 V_rms_ between a CLC cell (which is composed of chiral dopant S811 and cell gap is 10 µm) and a DFCLC cell. The optical signal of a typical CLC device was less frequency sensitive, as shown in Figure 2a. The electrohydrodynamic (EHD) effect causing the lower transmittance occurred under 20 kHz. Note that the temperature was measured by using a thermocouple. However, the dominant dielectric heating induced by the DHO effect (100–150 kHz) in DFCLCs exhibited dramatically lower optical transmission (FC and HFC states). In addition, an EHD also appeared when operated at a low-frequency electrical field (<20 kHz). This EHD instability can produce a rich variety of EHD patterns, causing the light transmittance to decrease slightly [15]. The obvious heating effect in the DFCLC material was observed, as shown in Figure 2a. Figure 2b depicts the electro-optical properties of a DFCLC cell between crossed polarizers under the applied frequency of 1 kHz. Figure 2c shows the frequency response transmittance at various applied voltages. Transmission as a function of the applied voltage field can also be exhibited in the DTSG. When the low-frequency (1 kHz) voltage field was applied, the DFCLC bulk showed positive dielectric anisotropy Δ*ε*, and the DTSG was initially in the P state. By increasing the applied voltage, the DTSG exhibited three main optical intensity states.

The highest transmission was the initial P state and the FC state, which appeared at ca. 15–70 V_rms_. Finally, the DTSG texture changed to a homeotropic (H) state when the applied voltage exceeded 70 V_rms_. However, the DTSG showed a negative dielectric anisotropic Δε when the applied voltage field was at the high frequency of ca. 100 kHz. The DTSG was initially in the cold P state when the frequency was below 100 kHz. When the applied frequency continued to increase, high-frequency voltage caused dielectric heating and increased *f_c_*. Finally, the dielectric anisotropic Δ*ε* changed to the positive dielectric anisotropic. The temperature dependence of the fc satisfies the Arrhenius equation, as shown in the expression:(1)$$f_{c}=A_{0}\exp\left(-\frac{E_{a}}{K_{B}T}\right)$$
where A_0_ stands for the material constant, *E_a_* denotes the activation energy, *K_B_* represents the Boltzmann constant, and *T* is the absolute temperature. Based on the equation, the *f_c_* increases with an increasing temperature. The dielectric heating power density in LCs can be expressed by [10]:(2)$$p_{d}=\frac{\omega \varepsilon_{0}E_{0}^{2}}{2}\frac{\left(\varepsilon_{s}-\varepsilon_{\infty }\right)\omega \tau }{\left(1+\omega^{2}\tau^{2}\right)}$$
where *ω* = 2π*f* represents the angular frequency, *ε*_0_ denotes the permittivity of free space, *ε_s_* and *ε*_∞_ stand for the static and high-frequency limiting dielectric constants, respectively, and *τ* is the relaxation time. Based on the power density equation, the power increases with the increasing amplitude of the electrical field. Based on these two equations, our experimental data were calculated and well understood. The texture of the dielectric heating-induced *f* < *f_c_* and Δ*ε* > 0 transforms the DTSG from the P state to the HFC or H state, as shown in Figure 2c. Further, Figure 2c shows the voltage-dependent DHO effect in a DTSG. The higher voltage amplitude yields a higher dielectric heating power, which results in a blue shift of the *f_c_*, and, in turn, produces a blue shift in the scattering HFC states. Figure 3a illustrates how the rise in temperature and the frequencies of appearance of the HFC/FC states in the DTSG varied with the applied voltage. The rise in temperature of the DTSG was linear, observed at applied voltages of 30–60 V_rms_. Further, the DTSG rise in temperature will increase rapidly when the applied voltage exceeds 60 V_rms,_ and slowly increase with the highest temperature when the voltage exceeds 70 V_rms_. Figure 3b demonstrates the light intensity of scattering HFC/FC state varied with the applied voltage at *f_c_*. Excellent performance of the DTSG is observed at an applied electric field of 60 V_rms_. In addition, the transmission spectra of tuning DTSG in heated mode from HFC to HP at 60 V_rms_ is shown in Figure 4. The lowest intensity of HFC state is at 145 kHz, and the highest intensity of HP state is at 165 kHz and 60 V_rms_. The contrast ratio of the DTSG device deduced from the transmittance ratio of HFC to HP with data retrieved from Figure 4 is 6–7. Compared with its well-known counterpart—LC smart glass—the most advantageous feature of this smart glass is the heating effect, whereby it takes a couple of seconds (~5 s) [7] to switch between cold mode (P and FC states) and heated mode (HP and HFC states). In the previous research, the stability of P and FC stable state in DTSG endured for several weeks [7]. Compared with the traditional heated glass technology, which is operated with high voltage (>100 V), our DTSG owns higher energy efficiency. Figure 4 shows photographs (on black paper) and optical textures of the DTSG device in the bistable HFC and HP states. It is worth mentioning that the HFC and HP states are bistable and have excellent scattering and transparent characteristics in the DTSG. If the voltage field is turned off, the HP and HFC states will gradually change to P and FC states due to the dielectric heating power vanishing, respectively. Compared with the other recent smart glass manipulating the light transmission properties via voltage, light, or heat, the proposed DTSG is the first glass can control light and generate heat at the same time.

### 3.2. DTSG Materials for Biosensors

LC biosensors were invented by Dr. Abbott in 2001 [16]. Since then, LC biosensors have become an important sensing technology. The LC biosensor technology uses biomolecules immobilized on a substrate to induce the vertical-to-planar reorientation of LC molecules to make a device with high sensitivity. However, this kind of LC biosensor lacks a heating-induced antibacterial property. Figure 5 shows the mechanisms of the DTSG biosensors with and without BSA biomolecules on the DMOAP layer. The immobilized biomolecules diminish the vertical alignment force of the DMOAP substrate. When the biomolecules BSA are adsorbed onto DMOAP-coated glass, the vertical anchoring power becomes much weaker. These BSA allow DTSG molecules to transfer to the P state. The DTSG transiting from the FC to P mode makes the Bragg reflection property more obvious. To prepare a DTSG device for detecting BSA, we immersed ITO glass substrates for 0.5 h in a fluid-form solution, including 1.6% (*v*/*v*) of DMOAP in Deionized water (DI) water to induce DMOAP-coated substrates. Prior to the BSA immobilization, 0.5–10 μg/mL of BSA solution was dispersed onto a DMOAP substrate for 30 min. Once rinsed with DI water to remove unbound BSA, the cells of each DTSG were self-assembled with a 10 μm spacer. The details of the BSA solution immobilization and device fabrication can be found in past publications [17,18]. The DFCLC changed to the P state with increasing concentrations of BSA. Ultimately, the qui-perfect P state led the DTSG biosensors to be in a reflective perfect P state (Figure 5) [17]. In addition, the DHO effect produced by the electrical features of DTSG can be applied to bioinspired sensing with regard to antibacterial surfaces. To investigate the voltage-induced heating property, a function generator was employed to supply square-wave voltages 70 V at 100 kHz on the DTSG biosensor to produce the HP of DFCLC through dielectric coupling DHO. The change of HP reflection light intensities could lead the DTSG biosensor to be used as a sensitive biosensor. Thus, increasing BSA concentrations could yield a higher optical intensity in the DTSG biosensor. The transmittance correlation and Bragg reflection of various BSA concentrations is also displayed in Figure 6. The experiment results show that the transmittance (Bragg reflection) of DTSG in the light spectrum could be used to detect and quantitate the biomolecules in this manner [18]. Therefore, this concludes that the BSA concentration cannot be detected by more than 8 μg/mL. Besides, the detection limit is about 0.5 μg/mL, because the transmission difference of DTSGs cannot be discerned beyond 0.5 μg/mL. The fitting relationship with a coefficient of R^2^ ≥ 0.92 was observed between 0.5 and 6 µg/mL of BSA (Figure 6). Therefore, the DTSG material could be used in biomedical sensing. The selectivity of LC biosensors has been demonstrated in the past [19,20,21]. Compared with the several based protein assays (absorbance- and fluorescence-based assays), the detection limit of BSA detected by our DTSG-based protein assay is more sensitive. Thus, with the proper voltage control, the DTSG device can be heated to have antibacterial efficacy (maximum temperature 60 °C), making the biosensors highly suitable for antibacterial deactivation applications.

## 4. Conclusions

The DTSG device based on a dielectric heating effect in tunable helical superstructure, CLC, is proposed. The electrical field-induced dielectric heating strongly affects the optical properties of LCs, which is known as the DHO effect. The DFCLC materials characterized by their special dielectric anisotropy and relaxation can be employed as a novel DTSG in heated and cold modes. Additionally, the DTSG can be driven by simply applying voltage-modulated frequencies. The DTSG possesses two transparent states (P and HP) and scattering states (FC and C). This novel DTSG offers excellent features and exhibits several potentials in temperature-dependent modulation, simple fabrication, and optical stability. Therefore, the DTSG material can be used in windshields, sunroofs, and heat preservation glass based on the special properties proposed in this study. The DTSG biosensor for the biomolecule detection with the antibacterial property was successfully established. Based on the results of the DTSG, we may be able to capture Respiratory Syndrome coronavirus 2 (SARS-CoV-2) and affect the arrangement of DTSGs. Combined with detection, this may have great advantages for coronavirus disease quarantine.

## Figures and Tables

**Figure 1 polymers-13-00245-f001:**
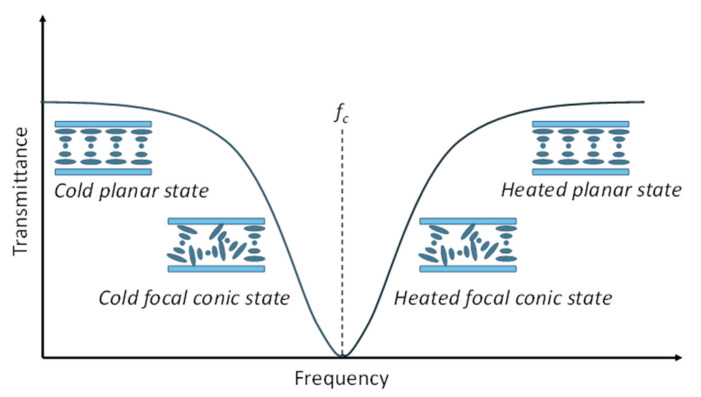
Schematics of the dielectric heating optical (DHO) effect behavior in the dielectric thermal smart glass (DTSG) device when varying the frequency of the AC voltage.

**Figure 2 polymers-13-00245-f002:**
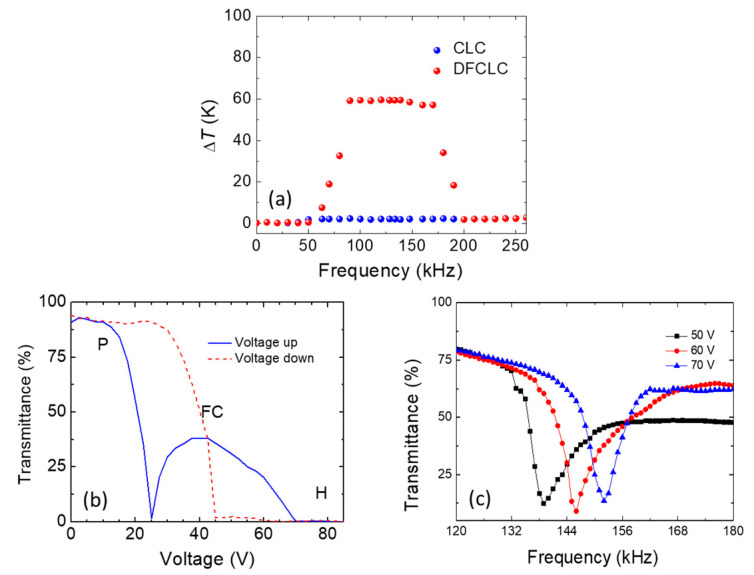
(**a**) The frequency-dependent temperature with various applied frequencies in the cholesteric liquid crystals (CLC) and dual-frequency cholesteric liquid crystals (DFCLC), (**b**) the voltage-dependent transmittance with three states, and (**c**) the transmittance intensity of heated focal conic/focal conic (HFC/FC) state varied with the applied voltage field.

**Figure 3 polymers-13-00245-f003:**
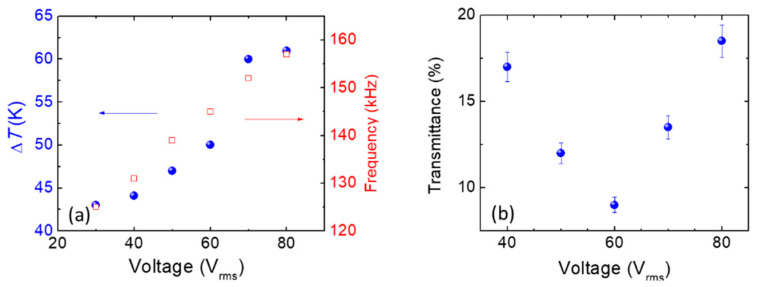
(**a**) The rise in temperature and the appearance of the HFC state in the DTSG by varying the frequency of the AC voltage, and (**b**) the transmittance intensity of the HFC/FC state varied with the applied voltage field.

**Figure 4 polymers-13-00245-f004:**
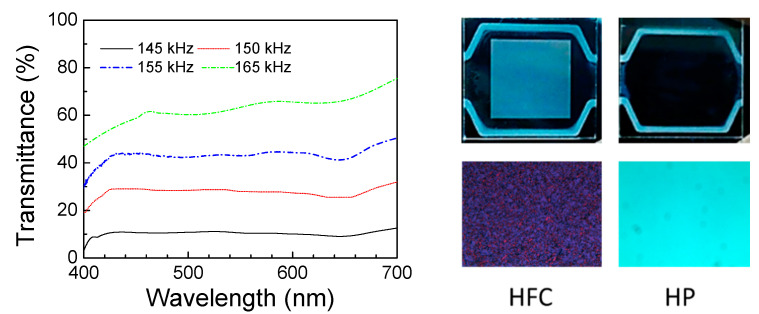
Transmission spectra of the DTSG in the HFC-to-heated planar (HP) state transitions under various applying electrical field conditions. Photographs of the DTSG device in the bistable HFC and HP states on black paper, and optical textures of the HFC and HP states under crossed polarizing microscopy.

**Figure 5 polymers-13-00245-f005:**
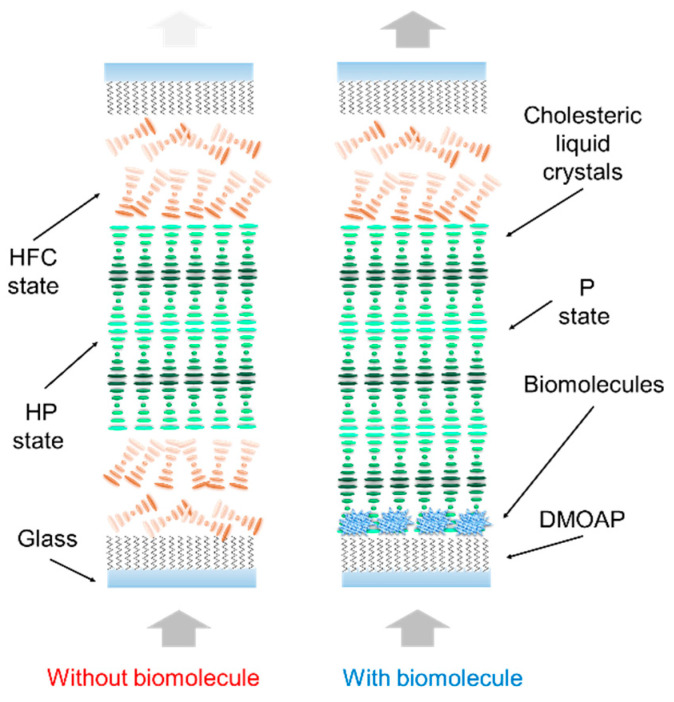
Schematic of the DTSG biosensor. The configuration changes from the HFC-HP-HFC to the focal conic-transparent planar (FC-P) mode in the presence of biomolecules on the DMOAP substrate.

**Figure 6 polymers-13-00245-f006:**
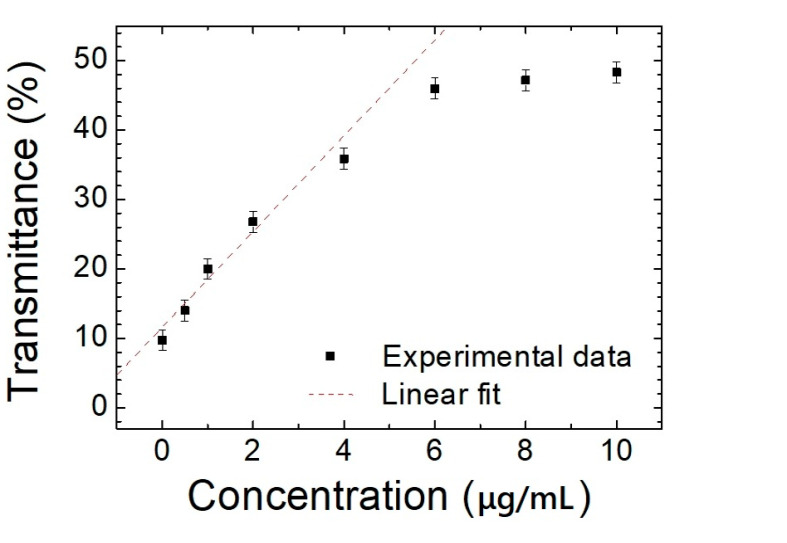
The correlations of the transmittance of DTSG at different BSA concentrations.

## Data Availability

The data presented in this study are available on request from the corresponding author.

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
