# Peer review of "Dielectric Thermal Smart Glass Based on Tunable Helical Polymer-Based Superstructure for Biosensor with Antibacterial Property"

_polymers, 2021, doi:10.3390/polym13020245_

Round 1

Reviewer 1 Report

The study presents the research results on ‘DTSG based on DHO effect in tunable helical polymer-based superstructures’. The originality of this manuscript should be emphasized more and this manuscript cannot be acceptable in its present form. Please consider the following comments and suggestion for further revision.

1. A concise abstract is required. Also, more specific descriptions of authors' finding should be added in the abstract rather than overall result of study. The abstract should state briefly the purpose of the research, the principal results and major conclusions.

2. There are many research results about dielectric thermal smart glass. What is the main advantages of this research? Please provide the advantage of these results and compare to other recent research papers. Please provide the novelty of this study and compare to other recent research papers in the view of technical points.

3. The explanation of results and discussion are not enough in ‘3.2. DTSG Materials forBiosensors’. Authors need to provide more detailed explanations and discussions. Regarding the results, it is necessary to present the comparison data with the recent research results related the design/assessment together with the relevant references.

4. For readers' understanding, it is necessary to describe the 'Schematic of the DTSG biosensor' in Figure 5 in more detail.

5. Authors should discuss the potential industrial application of this technology.

Reviewer 2 Report

Authors developed liquid crystals based optical biosensor. This topic is topics of interest for the Polymers and the manuscript is well organized. However, authors should test and confirm the performance and applicability of their biosensor. The modification points are as below:

  1. In this manuscript, there are so many abbreviations with similar spellings. For better readability, I would suggest to make a table with abbreviations. In addition, there is no explanation of DMOAP.
  2. There are two schematic diagrams. I would be better to combine and add clear principle of the signal generation. It is difficult to understand the signal generation mechanism for the biosensing. And mark the ‘P state’ in fig. 5.
  3. What is the molecular recognition layer of the biosensor? As I understand, authors simply immersed the substrate in the biomolecule solution. Then, it means that authors measured non-specific binding signal. In case of albumin, it can bind very well with non-specific binding. For the real application, biosensor should have molecular recognizing layer. If not, authors must show the selectivity of their biosensor by real-sample application (albumin spiked in human serum or other physiological fluid) and interference test.
  4. for better understanding, it will be better to add the table with the optimized parameters for the fabrication of biosensor.
  5. 0.92 of R square value is not sufficient to support the linearity (> 0.95 in general)
  6. More analysis for the performance of the biosensor should be added such as limit of detection, dynamic range, reproducibility, sensitivity, interference test and so on.

Round 2

Reviewer 2 Report

Authors answered according to the reviewers' comments. However, their response is NOT sufficient. They should give more detail and exact response of point #3, #5 and #6.

3. What is the molecular recognition layer? BSA is bind without any bioreceptor? Then authors should test or give exact data or reasonable reference. In addition, what is the final measuring solution of this biosensor, serum? Authors should show that their biosensor can recognized analyte in mixed solution. 

5. quasi-linear is not sufficient. What is the meaning of quasi-linear and why authors can say 0.92 of R square value is 'quasi-linear'?

6. How they calculate LOD and dynamic range? What is meaning of these value?

Round 3

Reviewer 2 Report

Authors reply according to the reviewer's comment so I recommend to publish this article in Polymers